# Fungal Biodeterioration and Preservation of Miniature Artworks

**DOI:** 10.3390/jof9111054

**Published:** 2023-10-27

**Authors:** Valeria Ventorino, Antonietta La Storia, Alessandro Robertiello, Silvia Corsi, Ida Romano, Luigi Sannino, Olimpia Pepe

**Affiliations:** 1Department of Agricultural Sciences, Division of Microbiology, University of Naples Federico II, 80055 Portici, Italy; valeria.ventorino@unina.it (V.V.); antonietta.lastoria@unina.it (A.L.S.); alessandro.robertiello@unina.it (A.R.); ida.romano@unina.it (I.R.); 2Museodivino, 80134 Naples, Italy; info@museodivinonapoli.it; 3SIB Società Italiana di Biotecnologie, 80055 Naples, Italy; ls@societaitalianabiotecnologie.it

**Keywords:** *Aspergillus*, miniature artworks, mold, antimicrobial activity, thyme essential oil, anaerobiosis, moisture

## Abstract

The study of biodeterioration is an important issue to allow the best conservation and prevent the decay of cultural heritage and artworks. In Naples (Italy), a particular museum (Museodivino) preserves the miniature artworks representing Dante’s Divine Comedy and Nativity scenes, executed with organic-based materials in walnut and clay shells. Since they showed putative signs of biodeterioration, the first aim of this study was to verify the presence of microbial colonization. A culture-dependent approach and molecular biology allowed us to isolate and identify the sole fungal strain Aspergillus NCCD (Nativity and Dante’s Divine Comedy) belonging to the A. sydowii sub-clade. Based on this result, a sustainable and eco-friendly approach was applied to find a method to preserve the miniature artwork by contrasting the growth of the strain NCCD. Several essential oils used as a natural biocide were tested against *Aspergillus* strain NCCD belonging to the *A. sydowii* subclade to determine their potential antimicrobial activity. Results revealed that basil, cloves, fennel, and thyme essential oils exerted antifungal activity, although their effect depended also on the concentration used. Moreover, anoxic treatment and the control of the relative humidity were used in the presence of thyme, in vitro, and in vivo assays to define the impact on fungal growth. No fungal development was detected in vivo in the shells treated with thyme essential oil at high relative humidity after 60 days of incubation at 28 °C. These results highlighted that although relative humidity was the major factor affecting the development of the strain *Aspergillus* NDDC, the application of thyme in an anaerobic environment is essential in contrasting the fungal growth. Identifying the biodeterioration agent allowed us to plan an eco-friendly, non-destructive approach to be successfully used to guarantee the conditions suitable for conserving miniature artwork.

## 1. Introduction

Biodeterioration is defined as an undesirable change in a material caused by the metabolic activities of organisms [1]. It is considered an important problem in the maintenance of cultural heritage because it represents diversified ecological niches in which various types of microorganisms could proliferate [2]. Many of the components of the works of art are biodegradable and consist of organic and inorganic compounds representing a substrate for the growth of microorganisms [3,4]. Among the components are glues, thickeners, emulsifiers, polysaccharides, proteins, oils, waxes, and gums [5]. The first phase of biodeterioration is favored by the interaction of the substrate, the environment, and organisms. The alterations are frequently studied at a macroscopic level and usually very little at a microscopic level to identify the microbial agents [6], a fundamental step to better understand the processes of biodeterioration of organic materials. For example, the organic material of the textile support of paintings on canvas is an excellent growth substrate for fungi and bacteria, and it spreads the microbiological alteration processes in different environmental conditions [7]. Concerning historical buildings, the colonization of external surfaces by microorganisms causes the deterioration of surface staining by biogenic pigments [8]. In addition, the pores are filled inside with the production of extracellular polymeric substances (EPS), creating mechanical stresses on the mineral structure and altering the size of the pore changes, humidity circulation, and temperature response [9]. The production of biofilm by the contaminating microorganisms leads to the formation of debris crusts on the rock surfaces due to the acidolytic and oxidation-reductive erosion of the mineral structure [10]. On the other hand, paper is certainly one of the oldest man-made materials, and the resulting biodeterioration of books and documents can damage the artistic and cultural heritage and bring considerable economic losses [11]. For the paper composition, it is necessary to evaluate the source of cellulose (wood pulp, linen rags, and cotton), coatings (natural/synthetic adhesives and starch), sizing compounds (gelatin, starch, and rosin), inorganic compounds (sulfates, carbonates, and clay), and contaminants (heavy metals) [12] that influence the attack of microorganisms, especially of fungi [13]. The principal components of wood are cellulose, hemicellulose, and lignin, and the microbial communities living in the environment can easily degrade them [14] by the production of extracellular and hydrolytic enzymes such as cellulases, chitinases, pectinases, pectinolytic enzymes, glycosyl hydrolases, and proteases [15,16]. Bacteria belonging to the genera *Arthrobacter*, *Bacillus*, *Microbacterium*, *Paucisalibacillus*, *Sporosarcina*, *Streptomyces*, and *Virgibacillus* [17,18,19], as well as fungi such as *Alternaria*, *Aspergillus*, *Aureobasidium*, *Cephalosporium* (=*Acremonium*), *Chaetonium*, *Cladosporium*, *Eurotium*, *Fusarium*, *Mucor*, *Penicillium*, and *Spicaria* [20,21], could grow on oil paints creating significant damage with the production of extracellular enzymes and acid metabolites [18]. Furthermore, the genera *Alternaria*, *Aspergillus*, *Chaetomium*, *Chrysonilia*, *Cladosporium*, *Drechslera*, *Exophiala*, *Eurotium*, *Fusarium*, *Mucor*, *Penicillium*, *Rhizopus*, *Rhodotorula*, *Trichoderma*, and *Verticillium* were found in wooden artworks [2,20].

The Campania region, in southern Italy, has an inestimable wealth of works of art [22], such as paintings and sculptures, to traditional works, such as Nativity scenes (also known as “presepe” in Italy). In particular, Museodivino (www.museodivinonapoli.it) in Naples’ old town, preserves small European miniatures of the twentieth century artwork created by Sacerdote Antonio Maria Esposito (SAME Collection). Scenes from Dante’s Divine Comedy set in walnuts and Nativities in little containers such as clay, chestnut, pistachio, cherry pit, hemp seed were created by the priest Antonio Maria Esposito between 1942 and 1999 using organic materials. The specific technique used was completely new in the history of sculpted miniatures. All artworks of the SAME collection were locked in small plexiglass boxes, which might have contributed to the state of preservation in time. In fact, during the summer of 2016, the biodeterioration of most of the “Dantesque walnuts” and some Nativities was observed by the appearance of several diffused yellow spots. Therefore, the aim of this research was to evaluate, isolate, and identify potential microorganisms that cause biodeterioration and to find a sustainable and eco-friendly approach to preserve the miniature artworks contrasting the growth of the microbial agents.

## 2. Materials and Methods

### 2.1. Artwork Description and Conservation Conditions

The SAME collection includes the Divine Comedy, which comprises forty-two walnut shells representing scenes from Dante’s work (thirty on Hell, eleven on Purgatory, and one on Paradise) and thirty-three Nativities created from the 1940s/1950s to the end of the 1990s. This is a unicum in the history of miniature sculpture with regard to the subject, composition, and materials used. Here, the characters, just a few millimeters high, are formed by little drops of oil paint, which were left to dry for months and then carved with surgical instruments, surmounted by tiny heads made with grains of pear pulp. The landscapes were usually entirely made with natural materials using pine buds, flower pistils, and moss. All the organic materials were immersed in turpentine.

In 2009, the shells were closed in transparent plexiglass boxes fixed with transparent silicone, leaving the back part free while completely closing the front part. From September 2016, the SAME collection was stored at Museodivino, built in a tufaceous basement, with a humidity rate between 70% and 75% and a constant temperature between 16 °C and 18 °C.

### 2.2. Sampling and Microbial Isolation

The sampling was conducted at the Department of Agricultural Sciences (Portici, Naples, Italy) from the fourth walnut shell containing the scene from Dante’s Divine Comedy and from the little clay shell with Nativities. The artworks were carefully extracted from plexiglass boxes in sterile conditions under a laminar flow biohazard cabinet.

Sampling was performed in sterile conditions using swabs from different points showing the same small yellow-orange stain present outside (NCDD1) and inside (NCDD2) the walnut shell representing Dante’s Divine Comedy, whereas putative microbial damage was present only inside (NCDD3) in the clay shell representing the Nativity. An example of a sampling point with putative microbial damage is shown in Figure 1. Samples were suspended in 5 mL of quarter-strength Ringer’s solution (Oxoid, Milan, Italy) and were left at room temperature for 10 min to allow the suspension of microbial cells in the solution. An aliquot from each sample was then streaked on plate count agar (PCA; Oxoid, Milan, Italy) to detect bacterial and fungal microbial populations and dichloran rose bengal chloramphenicol agar (DRBC; Oxoid) to isolate mold colonies (five plates for each medium) and incubated for 15 d at 28 °C. In addition, 1 mL from each microbial suspension was inoculated in tubes containing 10 mL of a plate count broth (PCB; Oxoid) enriched medium (two tubes for each sample) and incubated for 15 d at 28 °C. When the microbial growth was visible, the enrichment cultures were streaked on PCA and DRBC plates (five replicates for each medium) and incubated at 28 °C for 15 d. After incubation, single colonies were randomly isolated based on their colony morphology (i.e., shape, edge, color, elevation, and dimension) by repetitive streaking on the same isolation medium.

### 2.3. Phenotypic Characterization and Molecular Identification of Fungal Isolates

Microbial isolates were characterized based on colony morphology (shape, color, and dimension) as well as microscopic features (shape, dimension, and presence of spores) using the microscope Axiovert 200M (Zeiss, Milan, Italy) [22,23].

Mycelium was sampled after 48 h of growth on a PCA and DRBC solid medium, and genomic DNA was extracted and purified using InstaGene™ Matrix (Bio-Rad Laboratories, Milan, Italy) according to the supplier’s recommendations. Approximately 50 ng of DNA was used as a template for PCR assay. The synthetic oligonucleotide primers ITS1 (5′-TCC GTA GGT GAA CCT GCG G-3′) and ITS4 (5′-TCC TCC GCT TAT TGA TAT GC-3′) were used to amplify the ITS1-5.8S-ITS2 region [24]. The PCR mixture and conditions were prepared and conducted, as reported by Sannino et al. [25]. Amplicons were purified using a QIAquick Gel Extraction kit (Qiagen S.p.A., Milan, Italy), and eluted DNA was quantified and sequenced as previously reported [26]. Finally, nucleotide sequences were compared to the GenBank nucleotide data library using BLAST at the National Center of Biotechnology Information website (http://www.ncbi.nlm.nih.gov/Blast.cgi, accessed on 4 February 2022).

The primers V3f-GC and V3r and NL1-GC and LS2 were employed for bacterial and fungal analysis, respectively, using PCR mixtures and previously described conditions [27]. DNA extracted from the strains *Kosakonia pseudosacchari* TL13 [26] and *A. sydowii* VP4 [25] was used as a positive control for bacterial and fungal PCR, respectively. Water was used as a negative control. DGGE analyses were performed using a polyacrylamide gel on a Bio-Rad DCode Universal Mutation System (Bio-Rad Laboratories, Milan, Italy) with a denaturant gradient of 30–60% (run at 60 °C and 200 V for 240 min).

### 2.4. Approach to Preserve Miniature Artworks

#### 2.4.1. Antifungal Assay

The antifungal activity was evaluated in vitro using the agar well diffusion method [28]. The fungus isolated from shells was previously grown on malt extract agar (Oxoid) supplemented with chloramphenicol (100 mg L^−1^) to avoid the development of bacterial colonies [29] at 28 °C until spore formation. Conidia were harvested from the surface of plates by flooding the 10-day-old cultures with 9 mL of a sterilized quarter-strength Ringer solution and gently scraping with a sterilized glass rod. Then, 1 mL of a spore suspension, containing about 5 × 10^6^ spores mL^−1^, determined using the counting chamber Thoma (Hawksley, UK), was poured on malt extract agar plates in sterile conditions. Then, a hole was punched with sterile cork, and 5–10 μL of each essential oil diluted at 1:5 (*v*/*v*) with methanol was introduced into the well. Essential oils (Selerbe, BioDue S.p.A., Barberino Tavarnelle, Firenze, Italy) used in this study were basil, cloves, eucalyptus, fennel, lavender, pine and thyme. The plates were incubated for 7 days at 28 °C, the optimum temperature for fungal growth, to exclude a possible limiting temperature factor. Methanol was used as the negative control. The antifungal activity was evaluated by measuring the diameter of the inhibition area. Each experiment was carried out in three independent experimental replicates; the result is the average with standard deviation. The data were statistically analyzed by one-way ANOVA followed by Tukey’s HSD post hoc for a pairwise comparison of means (at *p* < 0.05) using the SPSS 21.0 software package.

#### 2.4.2. Susceptibility of a Biodeterioration Agent to Anaerobiosis, Moisture, and Thyme Essential Oil

Conidia was sampled after 48 h of growth of the fungal strain NDDC on a malt extract medium and suspended in 9 mL of quarter-strength Ringer’s solution (Oxoid, Milan, Italy) according to a microbial concentration of approximately 5 × 10^6^ CFU mL^−1^ determined using the counting chamber Thoma (Hawksley, UK). After shaking, tenfold serial dilutions (1:10) were performed and used to inoculate the malt extract medium. Plates were incubated at 28 °C under either aerobic or anaerobic conditions (Oxoid’s AnaerogenTM System, Oxoid, Milan, Italy) in a moist chamber (relative humidity 90%), and microbial growth was monitored at 5, 10, and 30 days of incubation.

To evaluate the effect of thyme, anaerobiosis, and moisture on the development of the biodeterioration agent in in vivo conditions, a set of walnut shells was created simulating the artworks of the SAME collection using the same materials that are to be used by the artist: droplets of oil paint for the realization of the bodies, grains of pear pulp for the heads of the characters, fragments of moss for the saplings, as well as the use of turpentine for the conservation of the perishable elements. Briefly, fragments of moss were immersed in the turpentine and positioned on the walnut shell using tweezers; then, the droplets of paint were applied by a thin brush, creating vertical shapes that recalled those of the characters’ bodies. Finally, pear pulp grains were immersed in the turpentine and positioned on the above painting characters. Walnut shells were then inoculated with fungal isolate. For inoculum preparation, conidia were harvested from the surface of malt extract agar plates by flooding the 10-day-old cultures with 9 mL of sterilized quarter-strength Ringer solution as described above. Inoculated walnuts were placed in gas-permeable polypropylene bags (SacO2, Deinze, Belgium) containing a 90 mm filter paper disc soaked with thyme essential oil diluted at 1:5 (*v*/*v*). The walnuts were incubated at 28 °C, the optimum temperature for fungal growth to exclude a possible limiting temperature factor, under aerobic or anaerobic conditions (Oxoid’s AnaerogenTM System, Oxoid, Milan, Italy), with two relative humidity levels (90% and 25%). Non-treated walnut shells were used as controls. Fungal growth on walnut shells was monitored for over 60 days of incubation, evaluating the development of fungal mycelium.

## 3. Results

### 3.1. Characterization and Identification of Biodeterioration Agents

Microbial colonies were obtained only from the enrichment PCB after 7 days of incubation at 28 °C by streaking on PCA and DRBC media. Isolates were identified as fungi based on their colony morphology. In detail, colonies were characterized by a mycelium white at the edges and yellow or green in the center, depending on the growth medium (PCA or DRBC, respectively), with a slightly elevated, and entire margin (Table 1).

All isolates observed under a bright field microscopy showed the same morphological characteristics. Vesicles were spherical to sub-spherical with 10–15 µm in diameter. Metulae and phialides were biseriate and covered half of the vesicle. Phialides grew on metulae and measured around 5.6–8 × 2–3.5 µm. Metulae measured around 6–7.5 × 2.5–4.2 5 µm. Conidia were globose to sub-globose and measured around 2.5–3.9 µm (Table 1). Hülle cells, specialized multinucleates as nurse cells, were found on hyphae during the sexual development phase and measured around 8–10 µm in diameter.

The sequence of the ITS1-5.8S-ITS2 region of all the fungal isolates showed a 99–100% identity with *Aspergillus tennesseensis* species using BLAST, strain NDDC (Nativity and Dante’s Divine Comedy).

Molecular characterization of bacterial and fungal community diversity colonizing shell samples was performed. DGGE analysis revealed that no bacterial strains colonized artworks since no bacterial bands were discovered, whereas only one band was revealed in fungal fingerprinting, confirming cultural analysis.

### 3.2. Approach to Preserve the Miniature Artworks in Shells

#### 3.2.1. In Vitro Antifungal Activity of Essential Oil

The antimicrobial activity of *Ocimum basilicum* L. (basil), *Eugenia caryophyllata* Th. (cloves), *Eucalyptus globulus* Lab. (eucalyptus), *Foeniculum vulgare* var. dulcis (fennel), *Lavandula angustifolia* L. (lavender), *Pinus sylvestris* L. (pine), and *Thymus vulgaris* L. (thyme) essential oils were tested using the agar well diffusion method, at different concentrations, against the *Aspergillus* strain NCCD belonging to the *A. sydowii* subclade. Results revealed that basil, cloves, fennel, and thyme essential oils exerted antifungal activity against the strain NDDC, although their effect also depended on the concentration used. In detail, as reported in Table 2, the highest significant inhibitory activity was observed with thyme at a dose of 10 μL, showing inhibition halos of 80.7 ± 1.4 mm, followed by a dose of 5 μL (64.4 ± 1.6 mm) and cloves (inhibition halos from about 47 to 68 mm). Basil and fennel essential oils exerted a similar antifungal activity at a dose of 10 μL (inhibition halo of about 30 mm), whereas, at a dose of 5 μL, an inhibition area of 28.9 ± 2.5 mm and 11.2 ± 1.8 mm was measured using basil and fennel essential oils, respectively. Finally, eucalyptus, lavender, and pine essential oils did not exert antifungal activity against the *Aspergillus* strain NCCD belonging to the *A. sydowii* subclade (Table 2). Figure 2 shows a representative image of the antimicrobial activity of *Eugenia caryophyllata* Th. (cloves) and *Thymus vulgaris* L. (thyme) essential oils that form the typical inhibition haloes against the *Aspergillus* strain NCCD belonging to the *A. sydowii* subclade in the solid growth substrate.

#### 3.2.2. Anaerobiosis, Moisture, and Essential Oil Inhibition Effectiveness in In Vitro and In Vivo Tests

To determine the efficacy of the effect of anaerobiosis and moisture on the fungal growth rate, an in vitro test was performed. Malt extract agar plates inoculated with the *Aspergillus* strain NDDC belonging to the *A. sydowii* subclade isolated from scenes of the Nativity and Dante’s Divine Comedy (NDDC) were incubated in aerobiosis and anaerobiosis conditions in a moist chamber. After 5 days of incubation, a concentration of 1.18 × 10^6^ CFU mL^−1^ was estimated in aerobic conditions, whereas no fungal growth was observed in the plates incubated in anaerobiosis and moisture. After that, three of the six plates where no microbial growth was detected were incubated in aerobic conditions to assess a putative fungicidal or fungistatic effect. After 10 days of incubation in aerobic conditions, a fungal growth (6.5 × 10^5^ CFU mL^−1^) was observed, a decrease of about 0.5 Log with respect to the control (1.18 × 10^6^ CFU mL^−1^). In the anaerobic plates, no microbial growth was detected yet. After 30 days of incubation, although microbial growth was found in anaerobic conditions, a reduction of about 1 Log was estimated in anaerobic conditions compared to the initial control (3.0 × 10^5^ CFU mL^−1^ vs. 1.18 × 10^6^ CFU mL^−1^).

To assess the combined effect of thyme essential oil with moisture in aerobic/anaerobic conditions, an in vivo test was developed. In detail, the strain NDDC was inoculated in the walnut shells created, simulating the artworks of the SAME collection, and treated with thyme essential oil. The shells were incubated under two different relative humidity (RH) levels (90% and 25%) in both aerobic and anaerobic conditions. After 60 days of incubation, no fungal growth was observed in the walnuts treated with thyme essential oil and incubated in anaerobic conditions, while microbial development was detected in the control walnut shells non-treated with thyme at 90% of RH (Table 3; Figure 3). In the presence of oxygen, strong colonization of walnut shells by the strain *Aspergillus* sp. NDDC was assessed at 90% of RH in both thyme-treated and non-treated samples, whereas no microbial growth was detected at 25% of RH (Table 3).

## 4. Discussion

The identification of microorganisms involved in the biodeterioration of cultural heritage is determinant in defining an efficient strategy to promote its preservation, especially for unicum artworks such as the walnut shells representing scenes from Dante’s Divine Comedy and the Nativity created by Don Antonio Maria Esposito stored at Museodivino (Naples, Italy). The three fungal isolates (NDDC_1_, NDDC_2,_ and NDDC_3_) recognized in this study as biodeterioration agents of the miniature artworks representing Dante’s Divine Comedy and the Nativity were identified using cultural characteristics, microscopic examination, and DNA sequencing, revealing that only one strain present was identified as *Aspergillus*, belonging to the *A. sydowii* subclade. To confirm this result, a molecular approach based on the use of DGGE was developed to assess the microbiota of the miniature artworks analyzed. Results revealed that only one fungal species was present in all of the artworks, whereas no bacteria were detected.

Among microorganisms having deteriorating effects on materials used for artworks, fungi are predominant in museums and warehouses all over the world, and therefore, they are considered among the most harmful organisms associated with the biodeterioration of organic and inorganic substances [2,30]. In fact, fungi can synthesize a wide variety of hydrolytic enzymes (e.g., cellulases, pectinases, chitinases, glycosyl hydrolases, proteases, and ligninase) for degrading organic matter [16]. Filamentous fungi of the genus *Aspergillus* are considered a threat to the conservation of cultural heritage objects because of their great enzymatic capacity and high biodeterioration power [31]. This genus comprises many species associated with the biodeterioration and alteration of a wide range of artworks such as books, manuscripts, paper materials, paintings, wood, buildings and stone artifacts, textiles, glass, human remains, and audiovisual materials [23,30,31,32]. Due to its metabolic versatility, these fungi were able to degrade recalcitrant polymeric materials such as lignin, cellulose, and hemicellulose [33]. Several *Aspergillus* species were previously isolated from wooden art objects and were identified as the dominant population able to colonize them, as well as the main alteration agent involved in the biodeterioration of these artworks [2,32,34,35,36,37].

Today, different strategies are applied to prevent microbial colonization of artworks. Among these, the use of natural biocide seems to be a promising approach. Although synthetic biocides have been applied to effectively eliminate a wide range of microorganisms and are especially used to control fungal spread and growth for many years, they could induce several negative effects on human and animal health as well as on the environment [38]. In this context, essential oils extracted from plants seem to be an eco-compatible and non-toxic alternative for the sustainable conservation of historic–artistic artifacts. Essential oils contain a wide variety of secondary metabolites able to act against several biological systems; therefore, they are thought of as environmentally suitable pesticides [32]. Essential oils are commonly used in food and pharmaceutical industries for their high antimicrobial activities against a wide range of prokaryotic and eukaryotic microorganisms, and only in the last decades have these been tested against the biodeterioration agents of artworks [39]. Although some essential oils showed low or no antimicrobial activity against the *Aspergillus* species [40], in the present study, the growth of the fungal strain NDDC was inhibited by thyme, basil, clove, and fennel essential oils also depending on their concentration used. This result was in accordance with Sparacello et al. [38], who reported that the growth of *Aspergillus* strains was inhibited by the application of essential oil extracted by *Thymus vulgaris* L. in an in vitro test. Recently, thyme essential oil was proposed as an alternative to commercial biocides due to its antimicrobial activity against the *Aspergillus* species in wooden artworks [32]. Moreover, Borrego and co-workers [41] used seven essential oils of plants as biocides against fungi and bacteria isolated from the National Archive of the Republic of Cuba and the Historical Archive of the Museum of La Plata, highlighting that clove was one the most effective against *Aspergillus* spp., *Fusarium* sp., and *Penicillium* sp. The antifungal effectiveness of essential oil obtained by basil was previously assessed by Fierascu et al. [42], demonstrating its potential against several fungal genera, such as *Aspergillus*, *Penicillium*, and *Mucor*, developed on paper artifacts. Moreover, Sanchis et al. [43] demonstrated that oregano and clove essential oils could represent a promising green strategy for the sustainable conservation of organic-based cultural assets against the *Aspergillus* species and *Trichoderma longibrachiatum* isolated from biodeteriorated archaeological mummified skin.

Since fungal contamination is determined by the availability of water, oxygen, and temperature, in the present work, the effect of anaerobiosis and humidity on the development of the fungal biodeterioration agent was evaluated in in vitro and in vivo conditions. Anaerobic conditions determined a growth reduction of the strain NDDC in in vitro tests. Recently, Boniek et al. [44] demonstrated that applying the non-destructive anoxic atmosphere technique determined the elimination of biodeterioration Aspergillus niger strains and their spores on a polychrome cotton painting. Recently, the anaerobic atmosphere technique and eco-friendly antifungal agents, such as essential oils of *Curcuma longa*, *Thymus vulgaris*, and *Melaleuca alternifolia*, were tested by Boniek et al. [45] against the biodeterioration fungal agents colonizing an engraving by Rembrandt, demonstrating that both nondestructive approaches were positive in contrasting the growth of the fungal strains.

The combined effect of anaerobic conditions, humidity levels, and thyme essential oil on the growth of the strain NDDC in walnut shells revealed that thyme is crucial in contrasting fungal development in anaerobic conditions at high RH (90%). By contrast, it seems that the treatment of thyme has no effect in the presence of oxygen, highlighting that the combination of aerobic conditions and high RH was the major factor affecting the development of the strain *Aspergillus* sp. NDDC in respect to an anoxic environment. Therefore, the environmental conditions, such as the rate of relative humidity (70–75%) and the low temperature (16–18 °C) established in the Museodivino may have determined and promoted the development of biodeterioration fungal strain. Kosel and co-workers [46] established that relative humidity was the key factor limiting the growth of different fungal biodeteriorating strains on wooden support, in which the fungal growth positively correlated to the relative humidity increasing (from 55 to 74%). Piñar et al. [47] reported that temperatures between 18 and 20 °C and humidity of 50–60% are recommended for the effective preservation of documents. Therefore, monitoring indoor conditions and controlling the museum’s micro-climate could be useful for addressing the restorers in applying the appropriate preventive measures related to the artwork preservation environment [48]. Automatic door-closers and air-exchange systems could avoid humidity fluctuations and optimize the control of temperature, resulting in decreased or ceased fungal growth [48].

In conclusion, this study proved that multiphasic, eco-friendly, non-destructive approaches could be successfully used without damaging the artworks during application for the preservation of miniature artworks of the SAME collection representing scenes from Dante’s Divine Comedy and the Nativity set in walnuts and clay shells affected by *Aspergillus* belonging to the *A. sydowii* subclade as well as to guarantee the conditions suitable for their conservation.

## Figures and Tables

**Figure 1 jof-09-01054-f001:**
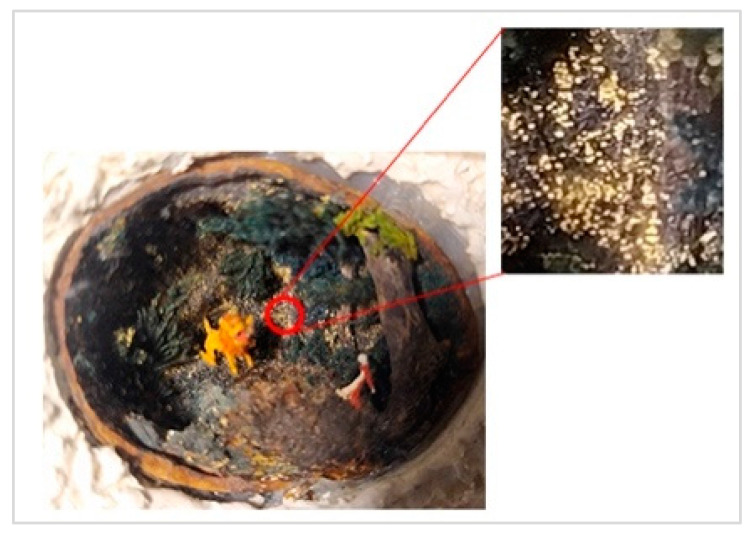
Sampling of yellow-orange point inside the walnut shell representing scenes of Dante’s Divine Comedy (4th NDDC; Inferno, Canto 1: The three beasts, vv. 31–60).

**Figure 2 jof-09-01054-f002:**
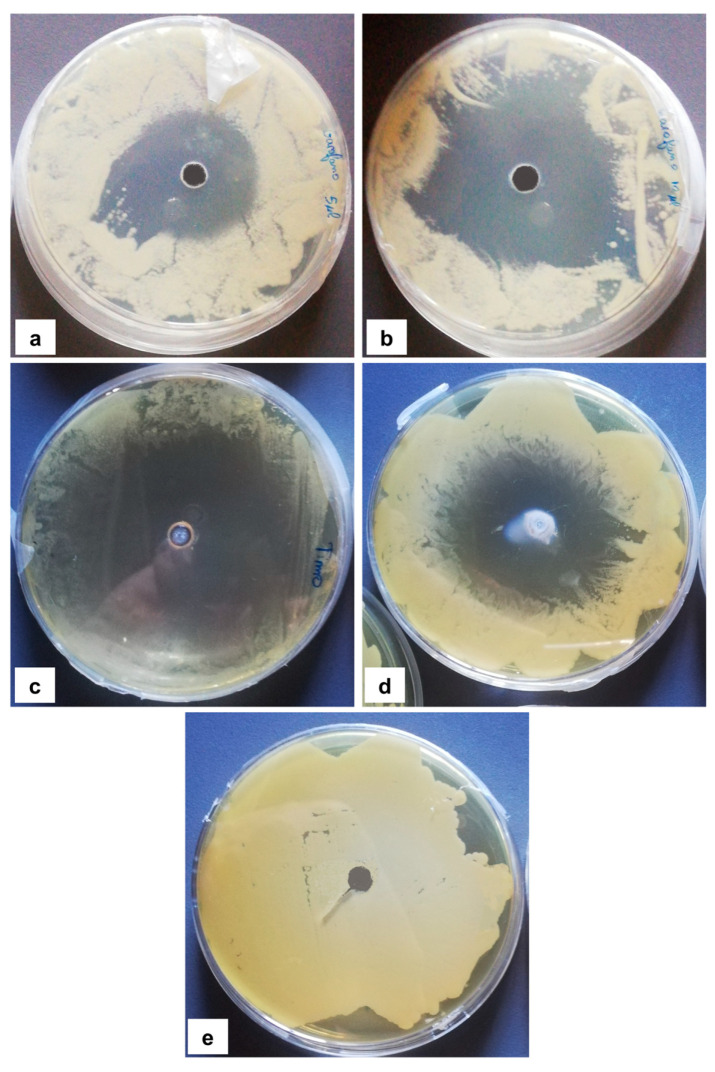
Representative antimicrobial activity of *Eugenia caryophyllata* Th. (cloves) (**a**,**b**) and *Thymus vulgaris* L. (thyme) (**c**,**d**) essential oils against the *Aspergillus* strain NDDC belonging to the *A. sydowii* subclade. The inhibition haloes were observed at doses of 5 μL (**a**,**c**) and 10 μL (**b**,**d**). Malt extract plates without essential oil were used as a negative control (**e**).

**Figure 3 jof-09-01054-f003:**
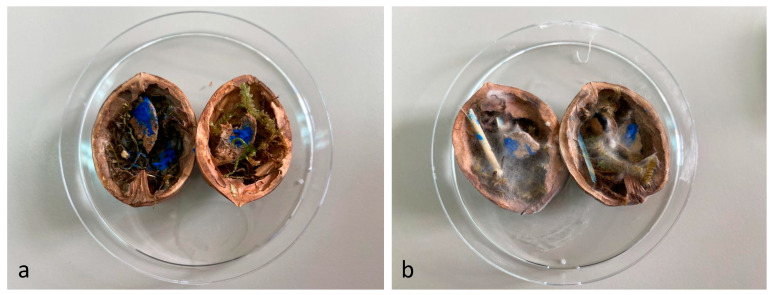
Walnut shells simulating the artworks of the SAME collection inoculated with the *Aspergillus* strain NDDC belonging to the *A. sydowii* subclade and treated (**a**) or non-treated (**b**) thyme essential oil after 60 days of incubation at 28 °C in anaerobic conditions at 90% of RH.

**Table 1 jof-09-01054-t001:** Colony morphology and micromorphology of fungal isolates from the outside (NDDC1) and inside (NDDC2) of the walnut shell representing Dante’s Divine Comedy and from the inside (NDDC3) of the clay shell representing the Nativity. The micrograph was obtained using the microscope Axiovert 200M with a magnification of 1000×.

Sampling Point	Colony Morphology		Colony Morphology Description	Microphotograph	Micromorphology Description
	DRBC	PCA			
NDDC1	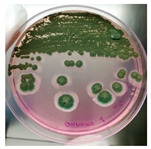	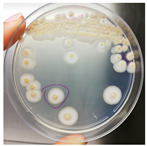	Mycelium white at the edges and green (DRBC) or yellow (PCA), slightly elevated, entire margin.	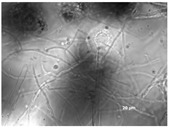	Vesicles spherical to sub-spherical, metulae and phialides biseriate, conidia globose to sub-globose.
NDDC2	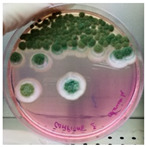	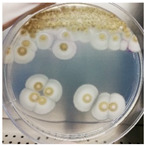	Mycelium white at the edges and green (DRBC) or yellow (PCA), slightly elevated, entire margin.	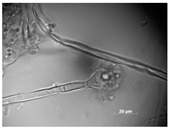	Vesicles spherical to sub-spherical, metulae and phialides biseriate, conidia globose to sub-globose.
NDDC3	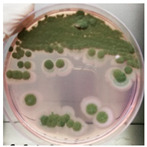	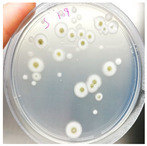	Mycelium white at the edges and green (DRBC) or yellow (PCA), slightly elevated, entire margin.	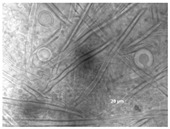	Vesicles spherical to sub-spherical, metulae and phialides biseriate, conidia globose to sub-globose.

**Table 2 jof-09-01054-t002:** Antifungal activity of different essential oils at different volumes (5 and 10 µL) evaluated as the diameter of the inhibition area (mm) against the *Aspergillus* strain NCCD belonging to the *A. sydowii* subclade.

Essential Oil	Inhibition Area (mm)	
	5 µL	10 µL
Basil	28.9 ± 2.5 ^d^	31.5 ± 3.4 ^d^
Cloves	47.0 ± 2.4 ^c^	67.6 ± 1.6 ^b^
Eucalyptus	n.a.	n.a.
Fennel	11.2 ± 1.8 ^e^	30.4 ± 2.7 ^d^
Lavender	n.a.	n.a.
Pine	n.a.	n.a.
Thyme	64.4 ± 1.6 ^b^	80.7 ± 1.4 ^a^

Data are means of three replicates ± SD. n.a.: not active. Different letters after values indicate significant differences (*p* < 0.05).

**Table 3 jof-09-01054-t003:** Growth of the *Aspergillus* strain NDDC belonging to the *A. sydowii* subclade on walnut shells treated and non-treated with thyme essential oil and incubated under two levels of relative humidity (RH; 90% and 25%) in aerobic or anaerobic conditions.

	Anaerobiosis		Aerobiosis	
	90% RH	25% RH	90% RH	25% RH
Presence of thyme	−	−	+	−
Absence of thyme	+	−	+	−

− no growth, + growth observed after 60 days of incubation at 28 °C.

## Data Availability

The obtained sequence in this study was deposited in the GenBank nucleotide database under accession number OP752154.

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
