# Peer review of "Fungal Biodeterioration and Preservation of Miniature Artworks"

_jof, 2023, doi:10.3390/jof9111054_

Round 1
Reviewer 1 Report
An enjoyable read, thank you. A nice introduction and explanation of this unusual project. Strengths of this work include the development of a novel in vivo microcosm recreating the composition of the artworks as a test bed, and the engaging discussion of similar projects undertaken to preserve various artifacts from fungal degradation. Weaknesses that need further explanation include the reasons for choosing the lab conditions that were used as they do not match the typical storage conditions of the artwork, and the controls used need to be properly explained.
Section 2.3 – what are the PCR and DGGE controls?
Line 170 - Why were the anti-fungal assays conducted at 28C when the fungus is growing in the artwork at 16-18C? Abiotic factors such as temperature can have an impact on fungal metabolism and their response to xenobiotic compounds. Please explain why the normal temperature that the fungus grows at (that the artwork is kept at) was not used for the experiment.
Line 183 – again, the same question. Why use 28C when the artwork is kept at 16-18C? How relevant is behaviour at 28C? Can you be sure that the same humidity and oxygen conditions would restrict fungal growth at a lower temperature?
Line 195 – what is the control for the thyme essential oil treatment?
Line 201 - It is not clear how the test humidities were chosen. 90% and 25% were trialled, but it is stated that around 70% is the normal humidity for the artworks to be stored in
Line 202 – how was fungal growth monitored?
Line 221 – it is remarkable that no bacteria were present on the shells. Were positive controls used to ensure primers and techniques worked properly?
Table 1 – the magnification of microscopy images should be included in the legend.
Lines 229-231, 234, 245 – the Latin binomial names of species should be italicized.
Figure 2 – this would be enhanced by the use of at least one further row of images showing thyme or other oil at two different volumes as a comparison.
Line 272 – how was the thyme oil applied to the walnut shells and their contents, and in what concentration?
Table 3 – how was growth observed? Visual or microscopic?
Overall, could this method be used without damaging the artworks during application of thyme oil?
The quality of English is acceptable but could be improved. The text is understandable as it is.
Author Response
We thank the reviewer for the work and comments to improve the manuscript that was changed as requested. Please, below you will find a list of point-by-point responses your comments and a description of the changes made that are highlighted in red in the revised version of the manuscript.
1. Section 2.3 – what are the PCR and DGGE controls?
- As reviewer suggested controls used in PCR for bacterial and fungal analysis were added in the revised version of the manuscript (lines 156-158: “DNA extracted from the strains Kosakonia pseudosacchari TL13 [26] and sydowii VP4 [25] was used as positive controls for bacterial and fungal PCR, respectively. Water was used as negative control.”).
2. Line 170 - Why were the anti-fungal assays conducted at 28C when the fungus is growing in the artwork at 16-18C? Abiotic factors such as temperature can have an impact on fungal metabolism and their response to xenobiotic compounds. Please explain why the normal temperature that the fungus grows at (that the artwork is kept at) was not used for the experiment.
- We thank the reviewer for the comment. The fungal strain is mesophile with optimum temperatures for growth between 25 and 30°C. Therefore, for its growth was chosen a temperature equal to 28°C to create the best conditions for fungal development and understand the effectiveness of the biodeterioration prevention methodology used even in a relatively short time of 60 days excluding limiting temperature factor. This was explained in the revised version of the manuscript (lines 174-176: “The plates were incubated for 7 days at 28 °C, optimum temperature for fungal growth to exclude a possible limiting temperature factor.”).
3. Line 183 – again, the same question. Why use 28C when the artwork is kept at 16-18C? How relevant is behaviour at 28C? Can you be sure that the same humidity and oxygen conditions would restrict fungal growth at a lower temperature?
- As above reported, the optimum temperature (28 °C) allows a faster growth of the fungal strain.In addition, the use of 90% of RH also allowed the fungi to be placed in an even more favorable environmental condition in combination with the optimal growth temperature.
4. Line 195 – what is the control for the thyme essential oil treatment?
- According to reviewer observation, control condition description was added to the text (line 210: “Non-treated walnut shells were used as controls.”).
5. Line 201 - It is not clear how the test humidities were chosen. 90% and 25% were trialled, but it is stated that around 70% is the normal humidity for the artworks to be stored in.
- We thank the reviewer for this comment. Fungal growth is favored by high humidity and therefore the use of 90% RH was chosen to place the fungal strain in an even more favorable condition in combination with the optimal growth temperature. Also the ability to growth under low humidity conditions (25%, no optimal value for fungal growth) was tested.
6. Line 202 – how was fungal growth monitored?
- As reviewer suggested more details were added to the text (lines 210-212: “Fungal growth on walnut shells was monitored for over 60 days of incubation evaluating the development of fungal mycelium.”).
7. Line 221 – it is remarkable that no bacteria were present on the shells. Were positive controls used to ensure primers and techniques worked properly?
- Yes, positive controls for bacterial and fungal PCR were used, as now described in the revised version of the manuscript (lines 156-158: “DNA extracted from the strains Kosakonia pseudosacchari TL13 [26] and A. sydowii VP4 [25] was used as positive controls for bacterial and fungal PCR, respectively. Water was used as negative control.”).
8. Table 1 – the magnification of microscopy images should be included in the legend.
- According to reviewer suggestion the magnification of microscopy images is included in the legend of the Table 1 (lines 240-241: “Micrograph was obtained by a microscope Axiovert 200M with a magnification of 1000X.”)
9. Lines 229-231, 234, 245 – the Latin binomial names of species should be italicized.
- We sorry for the mistake. According to reviewer suggestion all Latin binomial names of species are italicized in the revised vewrsion of the manuscript (now lines 246-262).
10. Figure 2 – this would be enhanced by the use of at least one further row of images showing thyme or other oil at two different volumes as a comparison.
- According to reviewer suggestion the Figure 2 was improved adding plates with thyme essential oil at the dose of 5 and 10 μL.
11. Line 272 – how was the thyme oil applied to the walnut shells and their contents, and in what concentration?
- According to reviewer observation, methods regarding thyme treatment was better explained in the revised version of the manuscript (lines 205-210: “Inoculated walnuts were placed in a gas permeable polypropylene bags (SacO2, Belgium) containing a 90-mm filter paper disc soaked with thyme essential oil essential diluted 1:5 (v/v). Walnuts were incubated at 28 °C, optimum temperature for fungal growth to exclude a possible limiting temperature factor, under aerobic or anaerobic conditions (Oxoid’s AnaerogenTM System, Oxoid), with two relative humidity levels (90% and 25%).”).
12. Table 3 – how was growth observed? Visual or microscopic?
- Visual, as now reported in the revised version of the manuscript (lines 210-212: “Fungal growth on walnut shells was monitored for over 60 days of incubation evaluating the development of fungal mycelium.”).
13. Overall, could this method be used without damaging the artworks during application of thyme oil?
- We thank the reviewer for this comment. Therefore a sentence was added in the conclusions (lines 401-405: “In conclusion, this study proved that multiphasic eco-friendly nondestructive approaches could be successfully used without damaging the artworks during application for the preservation of miniature artworks of the SAME collection representing scenes from Dante's Divine Comedy and Nativity set in walnuts and clay shells affected by A. tennesseensis as well as to guarantee the conditions suitable for their conservation.”).
Reviewer 2 Report
Interesting paper. Minor revisions are required, as marked in the attached PDF.

Author Response
We thank the reviewer for the work and comments to improve the manuscript that was changed as requested.
All correction suggested and reported by the reviewer in the pdf file are addressed and are highlighted in red in the revised version of the manuscript.
Round 2
Reviewer 1 Report
Acceptable responses and changes.
Author Response
We want thanks to the reviewer for comments to improve the manuscript that was changed as requested for publication in Journal of Fungi.